Using artificial intelligence to explore sound symbolic expressions of gender in American English

Kilpatrick Alexander alexander_kilpatrick@nucba.ac.jp 1
Ćwiek Aleksandra 2
1 International Communication, Nagoya University of Commerce and Business , Nagoya , Aichi , Japan
2 Leibniz-Zentrum Allgemeine Sprachwissenschaft , Berlin , Germany
Esuli Andrea
Electronic publication date: 2024 Jan 17
Publication date: 2024
Volume: 10
Electronic Location ID: e1811
Received 2023 Sep 28; Accepted 2023 Dec 18
Copyright: ©2024 Kilpatrick and Ćwiek
Copyright year: 2024
Copyright holder: Kilpatrick and Ćwiek
License: This is an open access article distributed under the terms of the Creative Commons Attribution License, which permits unrestricted use, distribution, reproduction and adaptation in any medium and for any purpose provided that it is properly attributed. For attribution, the original author(s), title, publication source (PeerJ Computer Science) and either DOI or URL of the article must be cited.
License URL: https://creativecommons.org/licenses/by/4.0/

Keywords: Sound symbolism, Gradient Boosting, Gender, American English

Funding: Japanese society for the promotion of science (JSPS) KAKENHI 20K13055 This work was supported by the Japanese society for the promotion of science (JSPS) KAKENHI Grant Number 20K13055. The funders had no role in study design, data collection and analysis, decision to publish, or preparation of the manuscript.

==============================
This study investigates the extent to which gender can be inferred from the phonemes that make up given names and words in American English. Two extreme gradient boosted algorithms were constructed to classify words according to gender, one using a list of the most common given names (N∼1,000) in North America and the other using the Glasgow Norms (N∼5,500), a corpus consisting of nouns, verbs, adjectives, and adverbs which have each been assigned a psycholinguistic score of how they are associated with male or female behaviour. Both models report significant findings, but the model constructed using given names achieves a greater accuracy despite being trained on a smaller dataset suggesting that gender is expressed more robustly in given names than in other word classes. Feature importance was examined to determine which features were contributing to the decision-making process. Feature importance scores revealed a general pattern across both models, but also show that not all word classes express gender the same way. Finally, the models were reconstructed and tested on the opposite dataset to determine whether they were useful in classifying opposite samples. The results showed that the models were not as accurate when classifying opposite samples, suggesting that they are more suited to classifying words of the same class.

Introduction

One of the central tenants of modern linguistics is that the sign is arbitrary (De Saussure, 1916). Human language is infinite in its ability to communicate because there is no logical or intrinsic relationship between the sounds that make up words and their meaning. For example, there is nothing particularly bee-like about the word “bee”. Or is there? What about the word “buzz”? In recent years, a growing number of studies have found systematic sound-symbolic patterns that challenge the arbitrariness of the sign (see Akita, 2015; Dingemanse et al., 2015; Nuckolls, 1999; Perniss, Thompson & Vigliocco, 2010; Kawahara, 2020 for a range of review articles from different perspectives). This field of research has come to be known as sound symbolism. In this study, machine learning algorithms are constructed to classify American English (hereafter: AmE) words according to gender. The purpose of this is twofold. Firstly, it investigates the extent to which gender can be inferred from the phonemes that make up words and given names through a machine-learning lens. Here, we use machine learning algorithms as very powerful statistical hypothesis tests which may allow for the detection of intricate data patterns that might prove challenging for established statistical hypothesis testing methods. Secondly, by using phonemes to train machine learning algorithms, we depart from traditional approaches to natural language processing. Traditionally, large language models are constructed using text-based data and rely heavily on lexical and syntactic features with little consideration to other elements of language. The general goal in natural language processing—a subfield of artificial intelligence—is to create algorithms that use and understand language in a similar manner to that of humans. By incorporating phonemes into the training of these models, we test the effectiveness of considering non-lexical features in natural language processing and incorporate elements of language that have previously been overlooked. The algorithms we used in our analysis are trained and tested on samples consisting of a gender classification as the dependent variable and a count of all possible phonemes in AmE as the independent variables (or features in the parlance of machine learning). Classification accuracy is examined to determine how much gender information is expressed in AmE and feature importance is analysed to examine which sounds carry gender information and how they contribute to a masculine or feminine classification.

One of the best-known cases of sound symbolism is the maluma/takete effect. First observed by Köhler (1929; 1947), and also known as the bouba/kiki effect, it is the observation that people will typically assign nonce words like bouba or maluma to rounded shapes and nonce words like kiki or takete to spiky shapes. In a study examining sound symbolic expressions of shape, Sidhu et al. (2021) examined 1,757 English nouns to explore the maluma/takete effect. They showed that the English lexicon carries sound symbolic information pertaining to shape where certain phonemes are associated with roundness while other phonemes are associated with spikiness. Those vowels typically associated with roundness (e.g., /u/ and /o/) are called rounded vowels because they are produced with lip rounding gesture, while those consonants typically associated with round objects (e.g., /b/ and /m/) are produced bilabially, that is by touching both lips together. This pattern is not apparently limited to English. Ćwiek et al. (2022) tested the maluma/takete effect on speakers of 25 languages from nine different language families. They found a robust effect in 17 of the 25 languages tested. More recently, Fort & Schwartz (2022) proposed that the maluma/takete effect might be explained by spectral balance and temporal continuity. In other words, a sound is perceived as round or spiky because it carries similar acoustic properties of the sound that a round or a spiky shaped object makes when it is hit against a hard surface.

Size is also robustly reflected sound symbolically in many of the world’s languages. In the context of vowels, this is often referred to as the mil/mal effect because words containing vowels like /i/ (e.g., mil) are often perceived as being smaller than words containing vowels like /a/ (e.g., mal) (Berlin, 2006; Newman, 1933; Shinohara & Kawahara, 2010; Ultan, 1978). For example, Shinohara & Kawahara (2010) tested speakers of Chinese, English, Japanese, and Korean to explore the judgement of size associated with vowels /a/, /e/, /i/, /o/, and /u/. They found that speakers of all languages judged /a/ to be larger than /i/. However, not all associations held across languages. While Chinese and Korean speakers judged /a/ to be larger than /o/, the opposite was true of Japanese speakers, and English speakers did not show a difference between the two vowels. This suggests that certain elements of sound symbolism are not universal but are specific to each language. For instance, in a study by Diffloth (1994), there is a case where /i/ is considered big and /a/ small, contrary to the typical pattern. Despite variations in how languages depict sound-symbolic contrasts, the evidence still supports the idea that these contrasts play a significant role, as iconicity remains integral to the linguistic structure. One explanation that relies on the physical properties of the sound source and exploits acoustic oppositions is Ohala (1994) “frequency code”. In both human and non-human species, vocalisation frequency is inversely correlated with size whereby larger entities typically produce vocalisations with lower fundamental frequency (F0). Low F0 can be considered a signal of threat or dominance in animal communication systems, and certain species have been shown to produce vocalisations with lower F0 as a deception mechanism to ward off potential threats (Morton, 1994; Bee, Perrill & Owen, 2000). An articulatory explanation for the mil/mal effect is that speakers are expressing the size of referents through the size of the oral aperture. Vowels that are produced with the tongue in a lower position (e.g., /a/ and /o/) create a larger space in the oral cavity while those produced with the tongue in a higher position (e.g., /i/ and /u/) create a smaller space (Whalen & Levitt, 1995).

Consonants have also been shown to carry sound symbolic information pertaining to size. For example, obstruent consonants produced with vocal fold vibration (e.g., /d/, /g/, /z/) are typically judged to be larger than those produced without vocal fold vibration (e.g., /t/, /k/, /s/) (Westbury et al., 2018). Experiments have shown this pattern to hold in the names of fictional video game characters in Brazilian Portuguese (Godoy et al., 2020), English (Kawahara & Breiss, 2021), and Japanese (Kawahara & Kumagai, 2021) where voiced obstruents tend to occur in the names of larger and stronger fictional characters known as Pokémon. Voicing on consonants is acoustically realized as low frequency energy and has been shown to increase the oral aperture, particularly the pharynx region, in an MRI experiment (Proctor, Shadle & Iskarous, 2010). In addition to voiced obstruents, nasal consonants (e.g., /m/ and /n/) have also been found to be overrepresented in the names of larger entities (e.g., Berlin, 2006). Nasal consonants are also associated with low frequency energy and may be associated with large entities because of the comparatively large size of the nasal cavity which is the resonance chamber for nasal consonants.

Languages can also carry sound symbolic information that is not related to the physical nature of referents. Abstract qualities such as rudeness (Aryani et al., 2018), humour (Westbury & Hollis, 2019; Dingemanse & Thompson, 2020), and politeness (Winter et al., 2021) have been found to be expressed sound symbolically. Adelman, Estes & Cossu (2018) showed that word initial phonemes in Dutch, English, German, Polish, and Spanish are significant predictors of emotional valence. Körner & Rummer (2022) report that the front high vowel /i/ is connected to more positive expressions, and /o/ and /u/ to more negative expressions in both German and Japanese, suggesting a cross-linguistic sound-symbolic effect. Many examples of abstract sound symbolism have come out of marketing research where sound symbolism has been used to explore branding strategies (e.g., Klink, 2000; Klink, 2001; Pathak et al., 2022). For example, Klink (2000) investigated word pairs with contrasting segments and their association with both physical and abstract qualities. They found that native AmE speakers perceived front vowels as not only smaller and faster, but also prettier, friendlier, and more feminine.

Previous studies have also explored how English expresses gender information sound symbolically. Some of these studies frame their hypotheses and explain their findings using the maluma/takete effect (Sidhu & Pexman, 2015) and the frequency code hypothesis (Pitcher, Mesoudi & McElligott, 2013). Sidhu & Pexman (2015), for example, showed that consonant phonemes typically associated with roundness were predictive of the female gender in names in Canadian English. On the other hand, Pitcher, Mesoudi & McElligott (2013) found that sounds with higher frequency were predictive of the female gender in AmE. The rational for using the maluma/takete effect as a basis for exploring sound symbolic expressions of gender is because women are typically more curvaceous than men, while the rational for using the frequency code hypothesis is because women are typically anatomically smaller than men (e.g., the total lung capacity is on average 1 litre less in females; Gick, Wilson & Derrick, 2013). These two hypotheses are incompatible, at least in the case of vowels in AmE, because one of the features of round vowels in AmE is that they all are produced at the back of the mouth. As above, rounded vowels have lower F2 than the unrounded counterparts (Stevens, 2000), and lower F2 is typically associated with increased size. In addition, lip rounding is known to lower the frequency of all formants because lip rounding gesture increases the length of the vocal tract and lip aperture (e.g., Smith et al., 2019). Consonants have also been shown to reflect gender sound symbolically. Sidhu, Vigliocco & Pexman (2022), showed that Canadian undergraduates associated sonorant consonants (e.g., /w/ and /l/) with femininity while both voiced and voiceless plosives (e.g., /t/ and /d/) were associated with masculinity. This finding is somewhat in contrast with Slepian & Galinsky (2016) who found that North American male names were more likely to begin with a voiced consonant (e.g., /d/ and /g/) while female names were more likely to begin with a voiceless consonant (e.g., /t/ and /k/) which is in line with the example from Klink (2000) explained in the previous paragraph. As noted earlier, some elements of sound symbolism are not cross-linguistic so these differences might be due to the different ways that Canadian English and AmE express gender sound symbolically.

In the present study, we construct extreme gradient boosted machine learning algorithms (XGBoost: Chen et al., 2015). The XGBoost algorithm is an advanced form of the random forest algorithm (Breiman, 2001) and was selected for this study because it was found to be slightly but significantly more accurate than the random forest algorithm in a similar experiment examining sound symbolism in Japanese (CH Ngai, AK Kilpatrick, 2023, unpublished data). In random forests, many decision trees are constructed using bootstrap aggregating (bagging: Breiman, 1996) and the random subspace method (Ho, 1998). Bagging involves randomly allocating samples to trees while the random subspace method involves randomly allocating features. By randomising across both dimensions, random forests are said to avoid overfitting. Decision trees in random forests are constructed independent of each other, so they do not learn from previous iterations. This is where random forest models and XGBoost models differ. XGBoost models construct sequential decision trees that take the results of earlier trees into consideration. Weak learners are trained on the residuals of stronger learners by focusing on areas in which earlier learners did poorly and increasing the importance of misclassified samples.

The XGBoost algorithms are trained to classify samples according to gender. The first model is constructed using given names and the typical gender of the referents. The second model is constructed using a list of nouns, verbs, adjectives, and adverbs which have each been assigned a psycholinguistic score of how they are associated with male or female behavior, among others (Scott et al., 2019). These models are trained and tested using different data subsets so that no samples involved in the training stage are included in the testing stage. Model accuracy is examined to determine whether and to what extend AmE communicates gender sound symbolically. Feature importance is investigated to ascertain which sounds are contributing to successful classification. Following this, we reconstruct each model using the entirety of each dataset and test it on the opposite samples. In other words, the model trained using names is tested using the list of nouns, verbs, adjectives, and adverbs and vice versa. We make the following predictions:

(H1) Both models will return a combined significant finding and an accuracy greater than chance, however we predict that the model trained and tested on given names will achieve a greater accuracy than the model trained on the Glasgow norms.

(H2) The feature importance of both models will show that low back vowels and voiced plosives will be associated with masculinity while high front vowels, voiceless fricatives, and sonorant consonants will be associated with femininity.

(H3) Of the models tested on their opposite dataset, both models will achieve an accuracy greater than chance, but both will also be less accurate than the earlier models suggesting that gender is reflected slightly differently in given names compared to other words.

Materials & Methods

All data and codes are available in the following repository: DOI 10.17605/OSF.IO/V46AD.

The XGBoost algorithms were constructed in the R environment (Build 548: R Core Team, 2021). The algorithms were constructed using the XGBoost package (version 1.5.0.2; Chen et al., 2015) and significance was calculated using Fisher’s combined probability test from the poolr package (version 1.1-1; Cinar & Viechtbauer, 2022). The hyperparameters for each algorithm were tuned by inputting various options into a tuning grid, so each algorithm was tuned to its specific dataset. The number of decision trees in each algorithm was set at 5,000 because a series of test models showed that stability and accuracy of each algorithm did not increase after 5,000 trees.

The data for the given names was taken from the Forebears website (Forebears, 2022) which lists the 1,000 most common names in American English. Transcriptions for the names were taken from the IPA-DICT project (ipa dict, 2022). Some names taken from the Forebears website were not present in the IPA-DICT corpus and were subsequently discarded from the analysis resulting in 989 names (female = 546). The IPA-DICT corpus provides a phonemic transcription for each name in the international phonetic alphabet (IPA); however, these were converted into ARPAbet because some IPA characters have functions in the R programming language. The data for the nouns, verbs, adjectives, and adverbs were taken from the Glasgow norms (Scott et al., 2019), a list of 5,553 English words which have been assigned Likert scale scores according to different psycholinguistic domains. The present study is concerned with the gender association domain which is described as how strongly a word’s meaning is associated with male or female behaviour. Each word was cross referenced in the Carnegie Mellon Pronouncing Dictionary (CMUdict: Weide, 1998) which provides a phonemic transcription in ARPAbet. Certain words in the Glasgow norms were not present in the CMUdict. These were excluded from the analysis resulting in 5,480 samples (female = 2,712). The two datasets differ in how they treat AmE mid-lax vowels. The CMUdict reports the mid-central, lax vowel, /ʌ/, but not the central mid-lax vowel /ə/, while the IPA-dict has central-mid, lax vowel /ə/, but not the mid-central, lax vowel, /ʌ/. Therefore, for the purpose of the analysis, /ʌ/ was converted to /ə/ so that the datasets had identical features which is important for when the algorithms are tested against their opposite dataset.

Samples in each dataset consist of the dependent variable and 39 independent variables. The dependent variable in both datasets is a categorical gender assignment. In the given name data, each name is assigned to either the male or female category according to a majority split to that gender as reported as a percentage on the Forebears website. For the Glasgow norms, gender classification was determined by a mean split according to Likert scale scores. The independent variables are all the sounds available in AmE. This results in a dataset that primarily consists of null values. For example, the name Chris, is transcribed as /kɹi s/ and is represented in the data set as a male classification with a score of 1 each for /k/, /ɹ/, /i/, and /s/, and a score of zero for the remaining 35 speech sounds. Null values made up 87.96% of the name data and 87.61% of the Glasgow norms data.

Having a dataset that is primarily made up of null values is problematic in decision tree-based algorithms because it undermines the effects of the random subspace method (Kilpatrick, Ćwiek & Kawahara, 2023). We addressed this issue by introducing another dimension for randomization: k-fold cross validation. In k-fold cross validation, the data is split into randomized folds which are then recombined to multiple testing and training subsets. In the present study, we use 8 folds (A-H). These are recombined to create subsets consisting of 2 and 6 folds whereby each iteration is trained using three quarters of the data and tested on the remaining quarter. For example, the first iteration of each model is trained on subsets A, B, C, D, E, and F, and tested on subsets G and H. There are 28 possible combinations of folds, so each model consists of 28 iterations. Given that each iteration constructs 5,000 decision trees, in total, 140,000 decision trees were constructed for each of the first two models. K-fold cross validation was not applied to the last two models because they are trained and tested on the entirety of each dataset.

Results

In line with H1, the algorithm constructed on the given name data achieved a higher classification accuracy (M = 67.33%, SD = 2.95%) than the algorithm constructed on the Glasgow norms (M = 58.55%, SD = 1.26%). Fisher’s combined p value calculations revealed both models to be significant (p = <0.001 in both cases). Despite achieving a higher accuracy, four of the 28 given name iterations did not achieve a significant classification accuracy while all the Glasgow norm iterations returned p < 0.001. Table 1 presents the combined confusion matrix for the given name algorithm and Table 2 presents the combined confusion matrix for the Glasgow norm algorithm. Interestingly here, is that the algorithm for the given names was much more accurate at classifying female samples while the Glasgow norm algorithm was fairly balanced in this regard.

Table 1 Confusion matrix for the algorithm trained and tested on given names.

		Classification	
		Male	Female	
Sample	Male	1,703 (24.6%)	1,398 (20.19%)	
Female	864 (12.48%)	2,958 (42.73%)	

Table 2 Confusion matrix for the algorithm trained and tested on the Glasgow norms.

		Classification	
		Male	Female	
Sample	Male	11,033 (28.75%)	7,951 (20.72%)	
Female	7,958 (20.73%)	11,439 (29.80%)	

To examine how the algorithms classify samples, we examine feature importance. This was calculated using the default method in the XGBoost package. This method assigns a score of 100 to the most important feature and a score to all other features that relativizes how important they are in comparison to the most important features. Table 3 presents the 15 most important features for the given names algorithm and Table 4 presents the 15 most important features for the Glasgow norms algorithm. In both tables, results have been aggregated. Gender allocation in these tables is calculated on average occurrence of each sound by gender and allocating each sound to the higher class.

Table 3 Combined feature importance for the given name model.

Only the fifteen most important features are presented.

Sound	Importance	Allocation	
/ə/	97.96	Feminine	
/i/	78.76	Feminine	
/n/	60.41	Feminine	
/ɹ/	34.07	Masculine	
/I/	29.24	Feminine	
/l/	27.55	Feminine	
/s/	26.41	Feminine	
/t/	23.03	Masculine	
/m/	21.25	Feminine	
/d/	21.03	Masculine	
/dɜ/	20.05	Feminine	
/k/	19.19	Masculine	
/b/	19.01	Masculine	
/æ/	18.41	Feminine	
/ʃ/	18.20	Feminine	

Table 4 Combined feature importance for the Glasgow norms model.

Only the fifteen most important features are presented.

Sound	Importance	Allocation	
/ə/	90.35	Feminine	
/ɹ/	83.49	Masculine	
/k/	80.15	Masculine	
/i/	80.13	Feminine	
/n/	75.95	Feminine	
/l/	75.34	Feminine	
/t/	75.15	Masculine	
/ʃ/	68.19	Feminine	
/p/	58.00	Feminine	
/ɛ/	52.15	Feminine	
/d/	50.32	Masculine	
/m/	44.13	Masculine	
/i/	42.58	Feminine	
/æ/	42.25	Masculine	
/ɝ/	42.08	Feminine	

In both models, non-back monophthongs were found to be highly important, but only really those that skew to the female gender. The mid-lax vowel, /ə/, was the most important feature in both models and it occurs more often in the female given names and in words with a feminine classification in the Glasgow norms. High front vowels, /i/ and /i/, are like the mid-lax vowel, being important in both models and occurring more often in words with a feminine association. The near-low front unrounded vowel, /æ/, presents an interesting case being important in both models. However, it occurs more often in female names, but more often in masculine words in the Glasgow norms suggesting perhaps that gender is reflected differently in proper nouns compared to other words in AmE. Diphthongs were not found to be particularly important in either model, though it is worth noting that the distribution of both monophthongs and diphthongs across genders followed a general pattern where low back vowels occurred more often in masculine words and high front vowels occurred more often in feminine words. Figure 1 presents a vowel chart that outlines the location and distribution of vowels.

Figure 1 Distribution of AmE vowels.

Monopthongs and dipthongs marked with an asterisk (*) had a distribution skew to masculine words in both datasets and those marked with a circumflex (ˆ) had a distribution skew to female words in both datasets.

The feature importance of vowels suggests that perhaps it is the frequency code rather than the maluma/takete effect driving gender-based sound symbolism. Unrounded high front vowels, which are characterised with spread lips and high F0 and F2, were found to be important in the classification of samples, while no rounded vowels were found to be particularly important in either model. Indeed, some of those vowels found to be important to the classification to the female category, namely high front vowels, are frequently cited as being used to represent spiky, rather than rounded, objects (e.g., Ćwiek et al., 2022). Those vowels that had a greater distribution to words classified as masculine were not found to be important to the algorithms. This suggested to us that perhaps vowels occur more frequently in female names than they do in male names. We conducted a count of the number of times vowels and consonants occur in each name and found that female names were made up of a greater percentage of vowels (M = 45.06%, SD = 9.86%) than male names (M = 39.89%, SD = 9.44%). We conducted a simple linear regression analysis to predict the percentage of vowels in names based on binary gender variables. It’s important to note that we opted for a linear regression model rather than a logistic regression, as our primary aim was to examine the relationship between gender and the proportion of vowels, rather than predict the gender itself. The regression equation yielded a significant result (t (1,987) = 8.859, p <0.001), indicating a statistically significant association between gender and vowel percentage, with an R 2 of 0.07.This effect was also found in the words from the Glasgow norms, albeit to a much lesser degree, where words associated with femaleness had a slightly greater percentage of vowels (M = 36.86%, SD = 9.41%) than words associated with maleness (M = 36.01%, SD = 9.49%). A second linear regression was calculated to test the Glasgow norms dataset, it also revealed a significant regression equation (t (1,5478) = 3.355, p < 0.001), with an R2 of 0.002. If an over representation of vowels in female names resulted on the algorithms emphasizing the importance of vowels, then the same might be said of consonants for male names.

To further interpret these findings, it is important to examine the effects of gender on the length of samples. Length is the summation of the number of phonemes in each sample. A linear regression analysis was conducted to investigate the relationship between gender and the length variable in the given names dataset. The model revealed a small but significant effect of gender on length, t (987) = 2.794, p = 0.005 with an R2 of 0.008. The estimated mean length for the feminine gender (M = 4.965, SD = 1.359) was 0.243 phonemes higher than that for the masculine gender (M = 4.722, SD = 1.359). A similar model constructed using the Glasgow Norms dataset revealed no significant influence of gender on word length using either the Likert scale values (p = 0.461) or the binary mean split values (p = 0.586).

In almost all cases, plosive consonants that were found to be important to the algorithms occurred more often in male samples. In the given names model, /t/, /d/, /k/, and /b/ were all found to be important to the model and all skew towards masculinity. This was true of /k/, and /d/ in the Glasgow norms model. The exception to this /p/ which was found to be important in the Glasgow norms model and skewed towards the female category. Cross-linguistically, including AmE, /p/ has been shown to carry sound symbolic information pertaining to friendliness or a lack of threat (Kilpatrick et al., 2023) which might account for its importance in the models. An alternate interpretation is that bilabial plosives are reflective of roundness according to the maluma/takete effect. Consonants associated with the female gender were the alveolar nasal, /n/, the lateral approximant, /l/, and the voiceless postalveolar fricative, /ʃ/. These were found to be important in both models and all three skew towards the female gender in both datasets. The bilabial nasal, /m/, was also found to be important in both models, however it only skews towards femininity in the given name dataset. Nasal consonants present an interesting case because nasal consonants carry low frequency so should be associated with increased size; however, /n/ was shown to reflect femininity in the present study while /m/ was ambiguous and has been shown to be associated with cuteness and softness in other studies (e.g., Kumagai, 2020). The only non-plosive consonant that was consistently associated with masculinity was the postalveolar approximant, /ɹ/, which was highly important in both models. Despite this, it is quite clear that sonority is linked to gender in AmE sound symbolism. Almost all plosives, that is speech sounds where airflow is completely obstructed, were associated with masculinity while almost all consonants that allow the passage of air were associated with femininity. This finding is in line with Sidhu, Vigliocco & Pexman (2022) findings who showed that sonorant consonants like /w/ and /l/ were associated with femininity while both voiced and voiceless plosives like /t/ and /d/ were associated with masculinity. Unlike Slepian and Galinsky (Slepian & Galinsky, 2016; see also De Klerk & Bosch, 1997), we found no evidence that contrastive voicing on plosives was suggestive of gender except in the case of /p/ in the given names and /b/ in the Glasgow Norms which were associated with femininity and masculinity respectively.

Given the similarities between the two models in terms of feature importance scores, it seemed that they would be useful in classifying their opposite samples. However, we observed a considerable drop in accuracy. For the algorithm trained using the given name data and tested on the Glasgow norms data, the model achieved an accuracy of just 53.16% (p < 0.001), and for the algorithm trained on the Glasgow norms data and tested on the given name data, the model achieved an accuracy of 57.63%.These findings suggest that there are ways that gender is expressed sound symbolically in AmE that are universal to both datasets, and there are ways that gender is expressed that are specific to each dataset. Table 5 presents a confusion matrix for the algorithm trained using the given names and tested on the Glasgow norms. Table 6 presents a confusion matrix for the algorithm trained using the Glasgow norms and tested on the given names.

Table 5 Confusion matrix for the algorithm trained using the given names and tested on the Glasgow norms.

		Classification	
		Male	Female	
Sample	Male	1,624 (29.64%)	1,144 (20.88%)	
Female	1,423 (25.97%)	1,289 (23.52%)	

Table 6 Confusion matrix for the algorithm trained using the Glasgow norms and tested on the given names.

		Classification	
		Male	Female	
Sample	Male	190 (19.21%)	253 (25.58%)	
Female	166 (16.78%)	380 (38.42%)	

Discussion

Gender classification based on linguistic cues has long been an area of interest in the field of linguistics. Previous research has shown that certain sounds and patterns of speech are associated with masculinity or femininity, and that these cues can be used to accurately classify individuals according to gender (Cassidy, Kelly & Sharoni, 1999). In the present study, we examined the behaviour of two supervised machine learning algorithms that have been trained to classify samples according to gender using the sounds that make up words. Our results showed that both models perform better than chance, but the model trained and tested on given names was the most accurate. This finding suggests that given names carry more gender information than other classes of words. To determine how the algorithms make decisions and which sounds express masculinity and femininity, we examined feature importance. In line with the seminal study by Cutler, McQueen & Robinson (1990), high front vowels were found more often in female names. Altogether, our analysis of feature importance shows that the mid lax vowel, /ə/, high front vowels and sonorant consonants like /l/ and /n/ are important to the classification of femininity. On the other hand, plosives like /t/ and /k/, as well as the post alveolar approximant, /ɹ/, are important to the classification of masculinity.

These findings are in line with the frequency code hypothesis (Ohala, 1994), which posits that low-frequency sounds, produced by larger organisms, are associated with dominance and threat, while high-frequency sounds, produced by smaller organisms, are associated with friendliness and lack of threat. Although frequency alone cannot fully explain the gender-specific sound patterns found in our study, it is likely that the frequency code effect plays a role in shaping the sound characteristics of male and female names. Our work supports previous findings reported by several studies. For example, Cutler, McQueen & Robinson (1990) report that female English names are more likely to contain /i/, while Cassidy, Kelly & Sharoni (1999) report a model classifying names by gender with 80% success rate, while human participants achieved 93% accuracy. Suire et al. (2019) found that male names contain more lower-frequency vowels, while female names contain more higher-frequency vowels (like /i/) in Oelkers (2004) reports similar tendencies for German. Furthermore, Pitcher, Mesoudi & McElligott (2013) report that female names contain more “small” vowels, while male names contain more “large” vowels, which they attribute to anatomical differences, like described in the frequency code (Ohala, 1994).

Vowels tend to be the focus of such investigations. We can measure the vocal frequencies of vowels and relate them to the resonance bodies that produced them. Unlike Modern English, which was the object of our study, Slavic or Romance languages still pertain on marking the gender in names and nouns. In Polish, for example, there are only a few borrowed female names that do not end with the vowel /a/. This pattern flies in the face of the frequency code hypothesis and illustrates how the sound symbolism of gender can be culturally specific. Similarly, gender was marked in Old English and began to decline in Middle English, between 11th and 15th century. However, we still use some of the names from old times, typically stemming from Greek or Latin, thus, the features marking gender, like female names endling with the vowel /a/ may still largely exist. In line with this reasoning, we show that, generally, American English female names contain more vowels than consonants and that vowels, alongside sonorant consonants, are particularly important for the classification to the female gender, while it is the consonants, plosives and the postalveolar approximant in particular, are markers of male gender in given names in AmE.

While there are markers of the male gender in both datasets, femaleness seems to be marked. This is supported by the fact that most of the important features identified by the XGboost model are associated with femininity. As is shown in Tables 3 and 4, 10 of the 15 most important features in the given names model and nine of the 15 most important features in the Glasgow norms model had a distribution skew towards the female gender. These findings suggest a societal tendency to emphasize and distinguish femininity in the naming conventions of AmE. This observation aligns with the notion that, within the cultural context of AmE, there may be social pressures to express and highlight gender roles (Prentice & Carranza, 2002; Eagly et al., 2020) through sound symbolism in names and other words. These results provide valuable insights into the cultural dynamics surrounding gender identity and the role of language in reflecting and perpetuating societal perceptions and expectations (Lewis & Lupyan, 2020). Another interpretation might be that name length—at least in the case of the given names—is influencing feature importance scores; however, the influence of gender on length was only small. A reviewer noted that many names have feminine endings for etymological reasons, such as -ia (e.g., Patricia, Julia), -Vtte (e.g., Charlotte, Jeanette), -tty (e.g., Betty, Patty), -Vlla (e.g., Ella, Estella), -elle (e.g., Michelle, Estelle), and final -a (e.g., Rhonda, Adriana). These endings often stem from feminine noun markers in languages like Greek, Latin, or others. While this systematic pattern does not alter the findings presented in this paper, it may provide additional context to help interpret some of the results.

The accuracy of the models reflect how robustly gender is expressed sound symbolically across the two datasets. Given that gender identity is one of the few reasonably accurate predictions that parents can make at the time when names are typically assigned, it is unsurprising that the model trained and tested on the given names performed more accurately than the model trained on other words. Sound symbolism is not known to have a very strong effect, despite its reliability, and a classification accuracy of almost 70% for the given name dataset was surprising because the algorithms had only phonemes to go by. For comparison, in a similar study using the random forest algorithm, Winter & Perlman (2021) showed algorithms constructed to classify adjectives related to size adjectives—and not the entire lexicon—did so with a 65.38% accuracy. While the model trained on the Glasgow Norms exhibited lower accuracy, it is noteworthy that both models demonstrated statistical significance. This is important in the context of Natural Language Processing which seeks to construct models that use and understand language the same way that humans do. This study marks a departure from conventional approaches to constructing Natural Language Processing models, which often rely heavily on text-based semantic and word-level data. The finding that a phoneme-based model can make reasonably accurate predictions means that existing and future large language models might be improved by taking segment-level data into consideration.

This study presents an XGBoost algorithm for classifying names into binary gender categories which raises important concerns considering the evolving societal understanding of gender. It is crucial to recognize that society is increasingly moving away from rigid binary categorizations and embracing a more inclusive and diverse understanding of gender identities. The use of a binary classification system for names fails to capture the complexity and fluidity of gender, and may reinforce outdated stereotypes and assumptions. This is not our intention. Gender is now recognized as a spectrum, encompassing a range of identities beyond just male and female. Therefore, employing an algorithm that categorizes names based on binary gender overlooks the lived experiences and self-identified genders of individuals. It is vital to promote research and develop algorithms that respect and reflect the nuanced understanding of gender to avoid perpetuating harmful biases and exclusionary practices. Future studies that explore this subject matter might employ a more sophisticated approach to gender classification. Such an investigation might possibly reveal interesting aspects of our evolving societal understanding of gender.

A reviewer raised the concern as to the use of personal names as stimuli for gender bias analysis. Unlike semantically void nonce words or fictional character names, personal names often carry cultural, religious, or etymological significance. For instance, names like “Mary” or “John” have roots in biblical characters, potentially influencing parental naming choices. The suggestion is made to consider alternative stimuli, such as fictional character names with no discernable etymology. Additionally, an experimental elicitation approach, involving the generation and evaluation of arbitrary names by native speakers, could be explored. Addressing this methodological concern could pave the way for future research in this area, ensuring a more nuanced understanding of gender expression without the potential confounding influence of culturally and etymologically loaded personal names.

Conclusions

This study details the construction and output of two machine learning algorithms that are designed to classify samples into binary gender categories. Samples consist of popular names in AmE and the Glasgow norms, a list of English words that have been assigned psycholinguistic scores. The classification accuracy scores reveal that—somewhat unsurprisingly—gender is more robustly reflected in the sounds that make up given names than in other word classes. The feature importance scores provide valuable insights into the specific cues that contribute to classification. They reveal a reasonably consistent pattern across the two models, showing that gender is expressed in names and other word classes in a similar way. High front vowels and sonorous consonants typically reflect femaleness in AmE while low back vowels and obstruents typically reflect maleness. Future research might delve deeper into exploring whether these patterns hold cross-linguistically. Overall, this study uses artificial intelligence to uncover the intricate relationship between gender and language, shedding light on the multifaceted ways in which gender is encoded in AmE.

Supplemental Information

Supplemental Information 1 Glasgow Norms code

Click here for additional data file.

Supplemental Information 2 Code for Given names - Gender updated to M/F

df Gender <  − ifelse(dfGender = = 1, “M”, “F”)

1 = Masculine, 2 = Feminine

Click here for additional data file.

Supplemental Information 3 Given Name data

Click here for additional data file.

Supplemental Information 4 Glasgow Norm data

Click here for additional data file.

Additional Information and Declarations

Competing Interests

Author Contributions

Data Availability

The authors declare there are no competing interests.

Alexander Kilpatrick conceived and designed the experiments, performed the experiments, analyzed the data, performed the computation work, prepared figures and/or tables, authored or reviewed drafts of the article, and approved the final draft.

Aleksandra Ćwiek conceived and designed the experiments, authored or reviewed drafts of the article, and approved the final draft.

The following information was supplied regarding data availability:

All data, code and associated .env files are available at OSF: Kilpatrick, Alexander, and Aleksandra Ćwiek. 2023. “AME Gender.” OSF. November 14. doi: 10.17605/OSF.IO/V46AD.

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
