# Peer review of "Using artificial intelligence to explore sound symbolic expressions of gender in American English"

_PeerJ Computer Science, doi:10.7717/peerj-cs.1811_

## Round 0.1 · original submission · Minor Revisions

All the reviewers agree on the quality of the paper, considering it worth publishing, pending the requested corrections.

In particular, one reviewer stressed how the list of personal names could have a cultural/etymologic ground that overcomes the phonetic aspects. The authors should investigate this aspect in the revision, extending the comparison (that is already present in the paper) between the two models.

Reviewer 1 ·

Basic reporting

This is a well-written paper on using machine learning to look at sound symbolism in proper names and at words rated on gender association. Although I have a couple of historical quibbles (see 'Additional comments') the review of previous work is solid.

The names do not seem to have been included among the on-line resources.

Experimental design

The question and methods are clear. Since many of the findings replicate earlier work using traditional regression techniques, it is not entirely clear what, if anything, the machine learning has added to our understanding.

Validity of the findings

The authors report robust effects for proper names (largely replicating earlier work), and weaker, but still statistically significant, effects for the words rated on gender association.

Additional comments

l. 58/59: Kohler never used 'baluba/takete' as claimed. He used 'baluma/takete' in the early printings of the first edition of his book, and 'maluma/takete' thereafter. It has been claimed (Footnote 1 in Westbury, Hollis, Sidhu, & Pexman, 2018) that 'baluma' was a typo because it appeared only in the early _printings_ of the first edition of his book.

Although I know these authors are not the first to do so, to me it seems historically inappropriate to refer to the effect as the 'bouba/kiki' effect. These terms were first used in Ramachadran & Hubbard, 2001. Naming the effect after their modern stimuli is a classic example of Stigler's law of eponymy, which states that no scientific finding is named after the person who actually discovered it. We should give credit where credit is due and call the effect the maluma/takete effect, as it was originally named by its actual discoverer, Köhler.

Ramachandran, V. S., & Hubbard, E. M. (2001). Synaesthesia--a window into perception, thought and language. Journal of consciousness studies, 8(12), 3-34.

l. 124 The term 'round-sounding' is not very helpful here. What phonemes are 'round-sounding'?

l. 274 Why do the authors write that "The feature importance of vowels supports the hypothesis that it is the frequency code rather than the bouba/kiki effect driving gender-based sound symbolism"? Notwithstanding the frequency differences, why do they here reject that the vowels themselves may carry some sound symbolic weight on this context?

The names do not seem to have been included among the on-line resources. The authors might discuss the fact that many name endings are feminine for etymological reasons: X-ia (e.g., Patricia, Julia, Gloria, Marcia, Felicia); X-Vtte [V = vowel](e.g., Charlotte, Jeanette, Suzette, Brigitte, and many more); X-tty (e.g., Betty, Patty, Kitty); X-Vlla (e.g., Ella, Estella, Isabella, Marcella); X-elle (Michelle, Estelle, Janelle, Annabelle, Rochelle); and final-a (e.g. Rhonda, Adriana, Tina, Etta, Christa, Marietta). Many of these derive from feminine noun markers in Greek, Latin, or other languages. Such systematicity does not alter the findings reported in this paper, but it may make some of them more explicable.

l. 285 It is not totally clear what 'simple linear regression' was used. Presumably it was a logistic regression to predict the gender of the name? This should be clarified.

·

Basic reporting

no comment

Experimental design

The only major concern I have is that personal names should not be used as stimuli for gender bias analysis. This is because personal names are not semantically void, unlike nonce words such as "bouba/kiki" or fictional character names. Many English names, such as "Mary" or "John", are from biblical characters, whose religious significance may motivate a parent to name their child after a certain character. (For example, a Catholic mother may name their daughter "Mary" because it is her saint or favorite biblical character) Many English names, including those present in the authors' list of 1000 most common names, have easily discernable etymological meanings: "Gloria", "Victoria", "Rex", "Flora", "Celeste", "Candy", "Krystal", and so on. Even when the etymology is not transparent, such as "Sophia" (from Greek sophia 'wisdom'), a parent may look up the etymology of a name before giving it to their newborn child, wishing their child to bear certain virtues like wisdom. It is not desirable to ignore these religious and etymological motivations and assume that the names come from nowhere; in fact, this goes against the assumption of the null hypothesis that the personal names are completely arbitrary strings of phonemes, since they are in fact not arbitrary (religion-motivated, etymology-motivated, etc.).

I suggest that the authors use a different set of stimuli, such as: fictional character names (in movies, books, cartoons, etc.) that are semantically opaque. An example would be the cartoon character "Mojo Jojo" in the series Powerpuff Girls whose name is a completely arbitrary string of sounds without any etymology. If it's difficult to gather character names this way, an experimental elicitation could also work: The authors could generate a list of arbitrary names and ask native speakers to judge them as male-sounding or female-sounding names and then run their statistical model on the male-classified and female-classified names.

Given that the manuscript can overcome this methodological hurdle, I find everything else about it very well written and consider it ready for publication.

Validity of the findings

no comment

·

Basic reporting

• The reporting is very clear, the paper is extremely well written, reads very easily and is easy to follow.
• The literature review is thorough, I am familiar with this field and did not notice anything missing. I felt the amount of literature review was also really good, not too much and not too little, and it was presented very well.
• Structure is clear and easy to follow.
• The tables were very good. I did notice however that the image resolution in Figure 1 was quite low. I was also wondering, if the journal allows colour, whether using colour to distinguish the male and female vowels might be easier on readers than using the asterisks, though I leave this to the discretion of the authors and editor.
• The raw data is supplied in a neatly organized OSF repository. I particularly commend the authors on their R code which is very well commented and easy to follow. There are however some small suggestions I can make for the OSF repository, listed below:
o I noticed that the repository is missing a ReadMe file. It’s fairly easy to work out what the different files are from looking at the R code, but perhaps adding a ReadMe explaining which data is in which files would be nice as well.
o I noticed as well that two of the files are in .xlsx format. I’m not sure how important those files are, but the .xlsx format is not as robust as .csv or .tsv, so maybe these could be converted to UTF-8 encoded .tsv files to ensure robustness for the future.
o The Given_Names.csv files seems to be missing a column telling you which given name is represented by each row. It’s not necessary for running the analyses, but it would be nice to know this.
• There was just one sentence, however, that I found a bit difficult to follow. It’s in lines 84-86, “However, it has also been put forward that some seemingly contradicting evidence do point to the general pattern that some characteristics are generally sound-symbolically marked using the existing oppositions of sounds”. I’m familiar with the study you cite here, and I’m guessing you mention it because Diffloth was talking about a language where /i/ is big and /a/ is small, instead of the other way around, but I guess what you want to say is that the contrast is still sound symbolically marked even if there can be cross-linguistic variation in how it is marked, because iconicity is after all a process of construal. It was quite difficult to get that from the sentence in lines 84-86, however, so maybe have a go at rewording it. Since you say in the previous sentence “certain elements of sound symbolism are not universal but specific to each language” you could even just give the example from Diffloth but then emphasize that even if we have different construals of the contrast, the sound symbolic contrast is still important in lots of languages. This was literally the only sentence in the entire manuscript though that I found hard to follow, so well done with everything else!

Experimental design

• Excellent, original research within the scope of the journal. This is the first time I have seen extreme gradient boosted algorithms applied to sound symbolism, it’s very exciting!
• The research questions and associated hypotheses are well-defined—see lines 169-177. As well as providing further confirmation of previous findings on the sound symbolism as gender, I thought that the comparison between the two different datasets (the given names, and the nouns), and in particular the idea of testing algorithms trained on one dataset on the other dataset was really novel and interesting. The findings show how sound symbolism is a highly specialized phenomenon. That is, that it’s not just distributed randomly throughout the lexicon, but that sound symbolism can be targeted to particular semantic domains (hence why the algorithm for given names was more successful than the algorithm for general nouns). I was actually going to suggest that the authors come back to this point a bit more in the discussion if space allows (e.g., around line 342 would I think be a good place to insert it), because the specialized nature of sound symbolism is I think not so well acknowledged, although one other study I would mention that also discusses this is Winter and Perlman (2021)—they find that size sound symbolism in English is restricted to size adjectives (rather than being a property of the entire lexicon).
• I am not familiar enough with the statistical methods used in the study to be able to comment on the technical standard of the analysis (hopefully another reviewer can provide more feedback here), but as a non-expert I will say that I particularly appreciated the authors explanation of the extreme gradient boosted algorithms in the introduction. They explain the method, including how and why it differs from other approaches, really well. And I had no trouble understanding the authors’ presentation of their methods and results, despite not being familiar with this area of machine learning, so well done!
• In terms of ethics, I thought it was very good that the authors included the paragraph beginning in line 405 of the discussion, in which they acknowledge the limitations of the current methods in forcing a binary gender distinction, and call for new methods in future studies that better capture the gradient experience of gender that better reflects people’s lived experiences. Again, the authors should be commended here.
• The methods are described in sufficient detail that the study is replicable, and the authors also provide their code in the osf which is well commented and easy-to-follow, so this would be very doable. They use open-source software too which is always a plus.
• I think that in the Results section, where the authors present the confusion matrix for the given names in Table 1, the authors should add a sentence commenting that (looking at the table) the model is much better at classifying female names than it is at classifying male names, and that the high accuracy of the model for given names primarily seems to be driven by the accurate classification of female (rather than male) names. I would also mention this again in the discussion, in the paragraph beginning at line 376 where the authors discuss how most of the important features used by the model were female features, as it is relevant here too.

Validity of the findings

• Underlying data is provided on the osf. The main findings are very robust. I did however have some questions with the follow-up analysis of the vowels in female names in lines 281 – 293. The authors note that only vowels more common in feminine names (and not vowels more common in masculine names) came up as important predictors in the algorithm. Their proposed explanation is that this effect is because vowels occur more frequently in female than in male names, and this is supported by counting the number of vowels in male versus female names in the dataset, and performing a simple linear regression to predict percentage of vowels based on gender. However, it does not seem like the authors controlled for the length of the names when they did this comparison. My intuition is that women’s names are often longer than men’s, so the effect they find could just be a by-product of length differences between names of different genders. In my opinion, I would think that this is just a further example of how femaleness seems to be marked in the sound symbolism of names, as they mention in the discussion. And in fact this seems to be quite common with sound symbolic effects in general—that one side of the effect is stronger than the other. For example, I think bouba associations are stronger than kiki ones (e.g. Ćwiek et al. 2021), and /i/ is small is a stronger effect than /a/ is big (e.g. Winter and Perlman 2021).
• The conclusions are well stated, linked to the original research questions and limited to the supporting results.

Additional comments

Minor comments:
• Line 301 says that /p/ is the only bilabial plosive found to be important in the models, but /b/ was important too
• In line 311, where it mentions the association of the rhotic with male names, which was an important predictor in both models, it might also be worth mentioning that rhotic sounds in general were found to be associated with ‘roughness’ in a large, cross-linguistic study (Winter et al. 2022). The finding was for trilled rs in particular, but in English the rs used to be trilled historically and I think the association has persisted even though the phonetics of the phoneme are no longer as rough sounding in English; see the discussion section of Winter et al. 2022.
• In line 318, it says that “we found no evidence that contrastive voicing on plosives was suggestive of gender”, but the models found that /b/ was associated with masculinity, and /p/ with femininity (albeit across different models). Did you just forget this here?
• Line 368 of the discussion, where you mention how in Polish female names primarily end in the vowel /a/. Do you think you should maybe comment here how this actually goes against the frequency code hypothesis and the pattern found for English here, showing how the sound symbolism of gender can be culturally specific.

Miscellaneous questions to the authors:
• This doesn’t necessarily need to be addressed in the paper, but is just something I’m curious about that, not being very familiar with AI. I wanted to ask, since the aim of natural language processing is, as you say, to create algorithms that use and understand language in a similar manner to humans, and then you say some of these sound symbolic patterns are only detectable with very powerful statistical hypothesis testing (more powerful than regular methods), does that mean you think that human brains are as powerful as AI in detecting these patterns? I suppose they would have more information to work with than just the training data that you give the machine, but I thought it was an interesting question.
• Related to the previous point, I found the study you mentioned by Cassidy et al. where they made a model to classify the gender of names, but then also had human participants performing the same task for comparison, really interesting as well. I was wondering if you would ever plan to have a human control group too (perhaps in follow-up studies)? I suppose for it to be more comparable to the machine, the human control group should not be familiar with the names, so this kind of setup might not work very well with common English names, but maybe if you had names from another culture and investigated how well English speakers (and/or speakers of other languages) could guess the genders of those names, compared to how well a machine learning algorithm can, that could be interesting.
• Another thing I was curious about, since you have information on the percentages for each gender the different names are associated with from the Forebears website, did you look at whether the algorithm actually performed worse on the more androgynous names (where the gender divide is closer to 50/50), versus with names that are very strongly associated with one gender over the other?
• I am curious about the study by Cassidy et al. 1999 (reference 56 in the running text; although I just noticed it is actually reference 57 in the bibliography so you may need to regenerate that). The accuracy reported for the classification algorithm in this study (80%) is even higher than in the current study. Do you have any idea why this is? Was this study also on English, or another language? Was it just a smaller dataset? I found it quite striking.

General comment: Overall, this paper was a pleasure to review! It’s in remarkably good shape for a first submission, with very few changes needed.

---

## Round 0.2 · accepted · Accept

All the reviewers of the previous round of revision checked this new version and found that it addresses their comments. They all agree that the paper can be accepted for publication.

Reviewer 1 ·

Basic reporting

This is a well-written paper on using machine learning to look at sound symbolism in proper names and at words rated on gender association. They find robust effects for the former, and weaker, but still statistically significant, effects for the latter.

Experimental design

No comment.

Validity of the findings

No comment.

Additional comments

The authors addressed all the points I raised in my first review and I am happy to recommend publication.

·

Basic reporting

The authors have addressed my concern sufficiently as a separate paragraph. I deem the revised manuscript sufficient for publication.

Experimental design

No additional comments.

Validity of the findings

No additional comments.

Additional comments

No additional comments.

·

Basic reporting

Very good. The authors have made changes to clarify the few areas in the manuscript where the meaning wasn't clear and the manuscript as it is now is very clear.

Experimental design

Excellent. The authors have made changes to address the small concern I had with the length of the names influencing the findings, and this is all accounted for now.

Validity of the findings

Excellent. No further changes needed.

Additional comments

Again, I want to applaud the authors for their excellent manuscript. I would also like to thank them for answering my miscellaneous/curiosity questions. I will look out for the papers that the authors mentioned in their responses and look forward to hearing more about their research in future publications :)